# Diamond for Electronics: Materials, Processing and Devices

**DOI:** 10.3390/ma14227081

**Published:** 2021-11-22

**Authors:** Daniel Araujo, Mariko Suzuki, Fernando Lloret, Gonzalo Alba, Pilar Villar

**Affiliations:** 1Department of Materials Science and Engineering, University of Cádiz, 11510 Puerto Real, Spain; mariko.suzuki@uca.es (M.S.); gonzalo.alba@uca.es (G.A.); pilar.villar@uca.es (P.V.); 2Department of Applied Physics, University of Cádiz, 11510 Puerto Real, Spain; fernando.lloret@uca.es

**Keywords:** diamond, MPCVD growth, power electronics, electron microscopy

## Abstract

Progress in power electronic devices is currently accepted through the use of wide bandgap materials (WBG). Among them, diamond is the material with the most promising characteristics in terms of breakdown voltage, on-resistance, thermal conductance, or carrier mobility. However, it is also the one with the greatest difficulties in carrying out the device technology as a result of its very high mechanical hardness and smaller size of substrates. As a result, diamond is still not considered a reference material for power electronic devices despite its superior Baliga’s figure of merit with respect to other WBG materials. This review paper will give a brief overview of some scientific and technological aspects related to the current state of the main diamond technology aspects. It will report the recent key issues related to crystal growth, characterization techniques, and, in particular, the importance of surface states aspects, fabrication processes, and device fabrication. Finally, the advantages and disadvantages of diamond devices with respect to other WBG materials are also discussed.

## 1. Motivations

In the 2020 and 2030 Climate-Energy Packages, the EU committed to lower greenhouse gas emissions by 20% with respect to 1990 and 55% by 2030 (very recent EU target) and to reach a share of renewables of 20% by 2020 and at least 27% by 2030. Today, there is a great concern about the conflict between energy and the environment. In this context, it is becoming usual to expect that an extraordinary increase of the use of electricity in the energy production, transport, and consumption will help a sustainable future. Enormous energy savings and exciting enhancements in quality of life will be enabled by new power electronics (PE) energy conversion systems. All energy-consuming devices, from pacemakers and home appliances to electric vehicles and industrial waste processing plants, will be affected. All alternative, sustainable and distributed energy (DE) sources, as well as energy storage systems, will be tied to the smart grid (SG) through swift and efficient PEs converters.

For competitive low-carbon renewable energy, transport energy, and smart grid applications, the impact of power electronics is striking where an optimized electrical energy conversion is demanded by the society. Approximately 30% of all electric generated power utilizes power electronics somewhere between the point of generation and its end use. Power electronics is used for more efficient transport, renewable energy production, and distribution including in highly efficient electricity distribution over long distances via high-voltage direct current power lines (HVDC) as well as in the better control of loads in switching power supplies and variable-speed drives for motors that drive fans, pumps, and compressors. By 2030, it is expected that perhaps as much as 80% of all electric power will use power electronics somewhere between generation and consumption. However, with the current state-of-the-art electric equipment, the transformation of the electrical energy occurs with significant losses (in the order of 10% from the source to the point of use) because available semiconductors are not ideal for high power.

The key to the efficient transmission and conversion of low-carbon electrical energy is the improvement of power electronic devices, which must be durable and reliable in high-power environments to eliminate the need for auxiliary systems for its use in transport (airplane, cars, etc.). “Green electronics”, i.e., highly efficient electronic devices, are crucially important for our future energy system. A first estimation gives a 75% reduction in losses, representing about a 10 MW energy saving on a 300 MW HVDC converter. The disruptive approach of the use of diamond in electronic devices can contribute knowledge, new approaches, innovative materials, and skills arising from the cross-fertilization with other WBG materials to yield more efficient and cost-competitive energy technologies.

## 2. Diamond Properties towards Electronic Applications

Silicon is a well-established semiconductor material that has addressed the requirements of energy conversion for more than 50 years. However, it is widely recognised (as shown in research roadmaps on power semiconductor devices [1]) that a real step-improvement in power electronics will be obtained by employing devices based on wide bandgap semiconductor materials. These materials have superior electrical characteristics for power devices when compared to silicon. Many grid applications use multilevel converters, with 3.3 kV (4.5 kV IGBT pressed (or stack) pack is commonly used fo HVDC and now 6.5 kV is also commercially available) silicon power semiconductors that need to be set in series to reach the required voltage level. Higher voltage capability devices are then highly required, 15 kV being a first step. To reach a high level current, paralleling of devices is required. Power electronic devices based on wide bandgap semiconductors are now resulting in substantial improvements in the performance of power electronics systems by offering higher blocking voltages, improved efficiency and reliability (higher performance/cost ratio), easier paralleling, and reduced thermal requirements, thus leading to the realization of more efficient green electronic systems.

Among wide bandgap semiconductors, diamond is considered to be the ultimate semiconductor for applications in high-power electronics due to its exceptional properties. Its dielectric breakdown strength is three times higher than in silicon carbide (SiC) and more than 30 times higher than in silicon (Si). In addition, unlike most other WBG semiconductors, the carrier mobility is very high for both carrier types, and the thermal conductivity is unsurpassed (see Table 1). In the latter, we indicate the commonly reported values of mobilities measured by Hall bars set-ups [2,3] on microwave-plasma-assisted chemical vapor deposition (MPCVD) layers. Note that some authors report much higher values when achieving the measurements with time of flight (TOF) set-ups on very high purity single-crystal diamonds. Values of electron and hole mobilities as high as 4500 and 3800 cm^2^/Vs have been reported [4]. Baliga’s figure of merit is oriented to static power losses. Another typical way to compare the power semiconductors is to draw the theoretical relationship between unipolar on-resistance versus the breakdown voltage of Schottky barrier diodes (SBDs). This is represented in Figure 1 for different semiconductors [5]. Power device engineering is interested in minimising the on-resistance for a given breakdown voltage capability. Diamond is the best candidate, and even Ga_2_O_3_, despite its lower carrier mobility, is better than SiC and GaN thanks to its high critical electric field.

An important aspect to rise is that Si power semiconductor switches used in 90% of the power applications market are metal-oxide semiconductor (MOS) gate-controlled devices (vertically diffused metal-oxide semiconductor, VDMOS, IGBT). Thyristor-based structures (integrated gate-commutated thyristors IGCT; gate turn-off thyristors, GTO) are still used for high-power high-voltage applications, mainly because of the lack of equivalent performance MOS controlled devices. This is something that diamond could solve considering its very high breakdown field above 10 MV/cm. In general, WBG semiconductors could open the way to novel concepts and applications in the high-voltage field such as electric transport and energy generation and distribution. Diamond, as a material with exceptional properties, could provide solutions to industry by providing diodes and transistors that withstand voltages above 10 kV, but competition with other materials, especially silicon carbide (SiC), and the intrinsic limitations of diamond (hardness, size of the substrate, etc.) require a great deal of effort to improve the performance, especially to reach high currents.

## 3. Diamond Growth: Substrates, Techniques, and Doping

### 3.1. Diamond Substrates

Homoepitaxially deposited diamond is carried out over commercially available diamond substrates. Diamond substrates can be synthetized by high-pressure high-temperature (HPHT) or chemical vapor deposition (CVD) techniques. Synthetic and natural diamonds are classified on the basis of their impurity concentrations. All natural and lab-grown diamonds contain some nitrogen (N) impurities [6]. When the N content is high (hundreds or thousands of ppm), so that it can be characterized by infrared (IR) spectroscopy, diamond is classified as Type I. These N atoms can be placed replacing carbon atoms in the lattice forming aggregated (Type Ia) or isolated from each other N atoms (Type Ib). Type Ia is also subdivided into IaA, with nitrogen atoms forming pairs, and IaB, with four nitrogen atoms symmetrically surrounding a vacancy in the diamond structure. On the other hand, when nitrogen content is low enough to not be detected by IR (<10^17^ cm^−3^), diamond is classified as Type II. Usually, the nitrogen reduction is a consequence of the incorporation of Boron impurities. When both impurities contents are very low, the substrate is named Type IIa, which corresponds to the purest diamond crystals. When the boron concentration increases so it is higher than nitrogen, the substrate is called Type IIb, and it corresponds to a p-type semiconductive diamond crystal.

Above the impurities content, there are two main restrictions related to the available substrate: the crystalline defects density and the substrate size. The HPHT process provides high-purity and low-defect-density substrates (<10^3^ cm^−2^ in Type IIa). However, the size is restricted to ≤1 cm^2^ by the technological limitations of the method. Moreover, the prize increases drastically when a high-purity diamond crystal is required. For this reason, Type Ib is the most usual diamond substrate for electronic applications, which means low cost and an acceptable dislocation density of about 10^5^ cm^−2^. Larger substrate sizes can be obtained by CVD growth [7]. The largest single-crystal diamond substrate has been reported to have a diameter of ~3.5-inches based on Ir/YSZ/Si [8]. ^®^ Other groups have reported 2-inch-scale substrates labs also using Ir hetereoepitaxy [9,10]. However, the crystalline quality is still worse than that of HPHT (commonly dislocation density of 10^7^ to 10^9^ cm^−2^ in heteroepitaxy growth) [11,12,13]. An alternative method to obtain large diamond substrates is growing on a mosaic configuration by CVD. This technique results in substrates sizes above 5 cm^2^ but the bonding boundaries of the mosaic are very strained and defective [14]. Thus, up to now, it has not been possible to obtain larger than 1 cm^2^ good-quality diamond substrates.

### 3.2. MPCVD Growth and Parameters

It was in the 1980s when, for the first time, cheaper synthetic and reproducible grown diamond was carried out using the CVD process [15]. In synthetic diamond growth, the extremely high surface energy of diamond, which reflects the strength of the bonds that must be broken to create a new surface, leads to inefficient wetting of substrates surfaces by growing diamond species. In addition, the complexity of the chemical reactions requires a highly advanced understanding of the technique. The CVD process is quite different from HPHT and the natural diamond formation. As its name implies, chemical vapor deposition involves chemical reaction inside a gas-phase as well as deposition onto a substrate surface. Growth of diamond films by CVD must be conducted under non-equilibrium conditions to avoid the more stable sp^2^ graphite phase. Furthermore, during the CVD process, hydrogen radicals (atomic hydrogen) must be present to, among other things, remove non-diamond carbon, including graphite, which is formed on the diamond surface. Methane is commonly used as the carbon source for CVD diamond growth. The process procedure with the whole range of selectable process parameters is shown in Figure 2 [16].

This sketch illustrates the various direct and indirect adjustable parameters. The first group shows the different selectable process gases that can be used for CVD. The second group reflects a selection of energy sources for the activation of the chemical process, followed by ensuring parameters. Below that, there is the substrate with the growing diamond layer. The principal chemical mechanism relevant to the growth of diamond from gaseous hydrogen and hydrocarbon species was presented in 1993 by Butler et al. [17], and the diamond CVD growth processes have been continuously developed since this time.

CVD englobes several growth methods based on the nature of the energy source. Among them, microwave-plasma-assisted CVD (MPCVD) has several advantages for growth of high-quality diamond yielding superior electric and optical properties. The main reason is that a stable plasma can be generated without any electrodes in the vicinity of the diamond growth plasma [18]. In general, growth mechanisms by MPCVD or HPHT have been extensively studied [19,20,21,22,23,24], and the models developed are well-known and accepted. These models are mainly based on the relative growth velocities of four low index crystal planes: {100}, {110}, {111}, and {113}. The value of the velocity ratios (the so-called growth parameters) describing the global morphology of the crystal during the growth process allows predicting their final shape. This final shape is given by the slowest rate among facets sharing an edge that limits the growth [15,17,18].

However, some critical issues that still exist for the overgrowth of the homoepitaxially MPCVD diamond layers must be resolved before CVD diamond films can be industrialized. Increasing the growth rate of diamonds, Rg, while keeping the crystal quality is one of the most important. This rate is highly dependent on the ratio of the methane flow to the total source gas flow, C_me_ = CH_4_/(H_2_ + CH_4_). When it is increased, the process achieves higher Rg. However, in the same proportion, the crystalline quality of the diamond layer grown tends to become poorer. The increase of carbon atoms in the gas mixture leads to a less effective sp^2^ etching process, carried out by the hydrogen. This results in an appreciable increase of the superficial roughness and defects by the formation of secondary-nucleated non-epitaxial crystallites and non-diamond phases. Indeed, most of the reported high-quality diamond (100) films were grown with C_me_ below 1.0%. Consequently, the typical Rg is lower than 1 µm/h [25,26,27,28,29]. In fact, the most appropriate C_me_s for the growth of atomically flat MPCVD diamond films are 0.05%, which result in Rg of the order of only 0.01 µm/h [27].

Nevertheless, this issue can be, at least, partially overcome using high microwave power. Teraji et al. reported high-quality (free-exciton recombination emissions in cathodoluminescence (CL) spectra at RT) homoepitaxial diamond films at higher Rg by using a MPCVD reactor with 3.8 kW of microwave power in addition to a relatively high C_me_ of 4.0% [30]. It was discovered that a kind of lateral growth was dominant even at such a high C_me_ when the high-power MPCVD was employed [31]. Reaching high Rg MPCVD processes is required for the realization of commercially available diamond-based electronic devices [32,33], but it is not the only issue to face.

### 3.3. Doping Issues

Another challenge of devices technology faced by novel semiconductors is the need for local doping of n-type and p-type layers, for instance, to build the n+ source and the p-well in a n-channel transistor. Something that seems to be easy in silicon, because of the existence of numerous low energy donors and acceptors atoms, is particularly complex with other semiconductors. The nearest to Si is SiC, even if high temperature implantation is needed to p-type doping. On the other side, it is difficult to p-type dope GaN and Ga_2_O_3_. Typically, in these materials, it is easier, or rather less difficult, to dope the semiconductor during the growth stage (bulk growth or epilayer growth) than by using local doping techniques such as diffusion or implantation.

Concerning diamond, boron and phosphorus are widely used as p- and n-type dopants of diamond semiconductors, respectively. p-doping is relatively easy during growth using boron atoms; the covalent radii of boron (0.088 nm) and carbon (0.077 nm) are close enough to allow the incorporation of boron as substitutional sites [34,35]. However, it needs an exclusive reactor as boron contaminates the entire gas system, and then the non-intentionally doping level is not controlled but in the range of 10^15^–10^16^ cm^−3^. If undoped or n-doping is needed, it should be grown in a different reactor even using different gas lines. On the other hand, phosphorous has a covalent radius of 0.117 nm that makes its incorporation difficult, this being more effective on (111)-oriented diamond substrates. In fact, the first phosphorous-doped {111}-oriented diamond was reported by Koizumi et al. at 1997 [36], whereas the first phosphorous doping of {100}-oriented diamond was not achieved until 2005 [37].

The doping level is also an important aspect due to the difficulty of reaching high doping levels and the dislocations that can be originated by such doping. Concerning the n-type, although first results were obtained more than one decade ago with the consecution of p–n diodes [38], the difficulty of introducing phosophorus in substitutional sites means that only some groups are able to grow such diamond [6,39]. For p-doping, the strain generated by such atoms’ incorporation in the diamond lattice introduces dislocations in the grown layers. Either critical thickness [40] resulting from the generated stress or dopant proximity effects [41,42] is the mechanism that can be responsible for the defects generation. Thus, growers should be very vigilant about adequate doping levels with the growth orientation and growth parameters to avoid the introduction of lattice defects.

Impurites such as phosphorous and boron atoms are easy to identify by cathodoluminescence. Doping atoms pine the Fermi level and incorporate either aceptors (by B atoms) or donors (by P atoms) levels in the bandgap. Thus, the doping increases to an impurity band merging with the valence band above the metallic transition. Dean et al. reported the first identification of bound excitons in (natural) diamond in 1965 [43]. The dependence of the isotopic boron-bound exciton of the host-lattice subsequently revealed the change in the diamond band gap based on the purity of its content of ^12^C or ^13^C [44,45,46], while the fine structures of boron-bound boron were observed for the first time [47]. In 1993, the effect of the boron concentration on the relative intensities of bound and free excitons in polycristalline CVD diamond was reported by Kawarada et al. [48] Today, several studies on highly boron-doped (near the metallic transition) diamond have been performed for both single and polycrystal [49]. On the other hand, clear donor characterestics of phosphorus were evidenced in 1997 [36,50]. Since then, many groups have observed the neutral phosphorus-bound exciton [51]. Experimental models have been developed to estimate the content for both impurities in low and high doping ranges [52,53,54].

### 3.4. Diamond Surface Roughness Effect

A tentative solution to the doping problems in diamond is surface transfer doping [55,56,57,58]. Hydrogen-terminated diamond exhibits p-type surface conductivity after its exposure to air. This process is closely related to the surface of the diamond as it relies on the hydrogen termination of the surface and the contact with a suitable electron-accepting medium. It is well known that surface/interface roughness scattering deals with a negatively impact carrier mobility in other material systems [59,60,61]. Contrary to what might be expected, the increasing of the surface roughness seems to enhance the hole-mobility in the bidimensional sub-surface of diamond [62]. The authors arributed this phenomenon to an increase in activation sites, which, in turn, led to an increase in carrier density rather than mobility. Other authors observed similar conductances increasing after roughening the diamond surfaces with ICP and RIE plasma etchings. Despite the lack of understanding, the authors considered that it may be linked to removal of surface defects during the plasma process. Indeed, the experimental results in hydrogen-terminated diamond field-effect transistors show that the diamond surface roughness significantly affects the carrier density dependence of the mobility [63]. Modeling FETs behavior showed that mobility drastically decreases with roughness when it is above 1 nm, and atomically flat surfaces are then highly desirable [64]. What is clear is that the roughness of the diamond surface has a great impact on the conductance of the surface. The reasons why this occurs, however, are still an open topic.

### 3.5. Alternative Growth Geometries

The cubic lattice of diamond is formed by two superimposed face centered cubic (FCC) lattices, with a/4 along each dimension displacement. This structure is anisotropic and results in high dependence with the plane of growth for growth rates, impurities incorporation, and surface passivation. This particularity has been used by authors for growth along unusual orientations for applications such as defects reductions [65,66] or doping optimization [67]. More recently, the anisotropy of diamond has been considered as an advantage for the design of three-dimensional architectures for devices that contributes to overcome the classical issues in diamond technology. Diamond overgrowth over patterned substrates can be predicted based on the growth parameters [68,69]. Thus, it is possible to define a specific lateral surface to be more suitable for the device applications, e.g., an orientation that maximizes the dopant incorporation. In fact, when diamond is homoepitaxially deposited on (100)-oriented patterned substrates using very low methane contents, growth rates along the <100> direction are very low, so lateral facets are maximized [70]. In addition, it is well-known that doping reduces the growth rates [71]. Consequently, it allows the growth of selectively thin doped layers with a high accuracy [72]. Obviously, there is a tendency of planarization during the deposition process, but this planarization can be extended by adapting the growth parameters in order to maximize the lateral sides of the structures. This technology is unequivocally aimed at circumventing the unsolved technological issues of diamond-based device manufacturing, such as the etchings, the doping incorporations, or the defect densities.

## 4. Structural Characterization Techniques

There are still several open questions concerning diamond that should be answered that are mandatory to manufacture feasible and reproducible devices. First, obtaining substrates of enough quality and low defect density to grow the required diamond structures on them for the device remains a challenge. Then, developing the growing conditions to obtain the desired doping levels (p- or n-type) without defects that could affect the performance of the device is the main concern of the diamond grower community. Issues such as dopant or defects (point defects, dislocations, and planar defects) distributions need to be controlled. Manufacturing the device also requires the capability of obtaining good ohmic and Schottky contacts and fully controlled diamond/diamond, diamond/dielectric, and diamond/metal interfaces, as some examples of technological challenges. Therefore, structural characterization is crucial to achieve all these technological targets. However, from the characterization point of view, some advantages of diamond may become a drawback to carry out some studies: for example, the high mechanical hardness makes difficult not only the sample preparation for transmission electron microscopy (TEM) but also cleaving the sample to make local analysis versus depth or to make laser mirrors. This is the main reason why not many TEM-related results for micro/nanostructural characterization are reported in the bibliography. The commercialization of the FIB-Dual Beam (focused ion beam coupled to a scanning electron microscope, SEM, column) 25 years ago now makes possible the sample preparation for TEM studies [73,74,75,76,77,78] of single-crystal diamond, although this equipment is still not extensively introduced in laboratories and, moreover, specific diamond TEM sample preparation issues such as amorphization or redeposition during etching have to be fully controlled to be successful in further TEM observations. Therefore, other techniques have been more extensively used for diamond characterization, as Raman or FTIR spectroscopies, X-ray diffraction (XRD), atomic force microscopy (AFM), or even X-ray photoelectron spectrometry (XPS).

This last one provides useful information related to the chemistry, composition, and electronic phenomena of diamond interfaces. The measured kinetic energy of escaping electrons when the material is irradiated by an X-ray beam is mostly dependent on its original core-level energy, which allows its identification. The depth sensitivity of the technique is dependent on the inelastic mean free path, which, in turn, is a function of the kinetic energy of escaping electron and the nature of the material through which it travels. For diamond C 1s electrons excited by an Al-kα source (hν = 1486.6 eV), the inelastic mean free path has been experimentally estimated as ~2.4 nm [79], which gives a maximum depth sensitivity of ~10 nm. This very short sensitivity has promoted XPS for diamond surface termination characterization, which is a topic of great importance in further devices manufacture. However, under the mentioned XPS conditions, the intensity of the diamond surface contributions is relatively much lower than that of the bulk contribution, harming the spectra analysis and peak identification. To overcome this issue and obtain even more surface specific information, some authors have opted for two different approaches: the use of synchrotron or other tunable X-ray beam energy systems [80] or the use of angle-resolved XPS (ARXPS) mode [81]. In both cases, the number of studies is very low in comparison to those based on energy-fixed and angle-fixed XPS conditions while being the key for a better comprehension of diamond surface phenomena. Recently, the ARXPS mode has allowed the identification and reinterpretation of a surface downward band bending component on (100)-H-terminated surfaces [82,83]. It is remarkable to keep in mind that the use of adequate and precise models to interpret the material is of maximum importance to correctly analyze XPS peaks obtaining a good fitting with the experimental results. Figure 3 shows, as an example, how the recorded XPS C 1s peak has been decomposed based on a model consisting of a diamond bulk region (Peak_bulk_), a diamond with a downward band bending region (Peak_bb_), and one monolayer of C-H (Peak_CH_) XPS contribution for an H-terminated diamond surface.

On the other hand, the XPS results on O-terminated surfaces have allowed the attribution of different oxygenated species contributions such as C-O-C bridges, ketones, or hydroxyl [84]. However, due to the short energy distance among some of them and the wide oxygenation methods, there is still no wide agreement on the models of O-terminated surface. In this sense, the ARXPS detection and quantification of a reproducible carbon sp^2^ surface contribution could provide a turn to the current vision of (100)-O-terminated reconstructions models [85].

Concerning diamond junctions’ interface electronic phenomena, XPS can be used for the estimation of Schottky barrier height (SBH) in metal–diamond junctions [86,87,88], as well as band-offset in heterojunctions [89,90,91]. The estimation of the SBH is complementary to the electrical characterization, but its comparison gives an idea of the homogeneity of the contact throughout the contact area. Localized SBH variations could have a critical effect on the I/V performance while remaining negligible in XPS experiments. Thus, it is expected that the XPS method overestimates the SBH in comparison to the electrical characterization. Concerning heterojunction band-offset estimation, XPS is the main experimental method for this purpose. For the estimation of such parameters, the simultaneous XPS detection (within few nanometers) of diamond and the deposited material is required.

Even though the density of dislocations depends on the supplier, the most common ones have still a high density of such defects in addition to the presence of growth sectors where the incorporation of impurities can vary from one to another. The makes it difficult to make reliable electrical characterizations. Optical-related characterization can also deliver informations on such defects, but the wide bandgap makes an excitation above the bandgap energy difficult. In consequence, cathodoluminescence (CL) is particularly well-adapted for the diamond crystal characterization [30,31,32]. Its indirect bandgap favors the defect-related transitions. Exciton-related transition in the UV range gives important information either on the quality of the diamond crystal or on the dopant present in it. In the optical-range extended defects related to the A-band, point defects are well-known from published studies and charts [92]. During the last two decades, CL has been demonstrated [93] to be an exceptional tool to evaluate the doping level [94] with an accuracy one order of magnitude better than secondary ion mass spectroscopy (SIMS). It also allows determining the type of point defects present in the crystal as H3 that are generated in the mid-gap [95]. Dislocations related to mid-gap levels (A-band) can be related to the presence of interfaces using monochromatic luminescence maps with a sub-micrometric resolution. In addition to CL, carrier densities can be evaluated using optical infrared (IR) spectroscopy (reflection as well as transmission, where peculiar FIB designed geometries can be used, Fourier transformed IR, FTIR). Their relative peak intensities related to the valence band to impurity levels (or band for high doping) permit the deduction of the boron doping level. FTIR and CL both allow the determination of the active boron density (or p-type dopant level), while SIMS gives the total density of boron atoms present in the crystal [96]. Thus, these are complementary techniques. Concerning their spatial resolution, CL is clearly the most efficient one with a resolution below the µm when used on FIB-prepared sample, which allows avoiding the pear-shaped volume of interaction between the high energetic electron beam and the diamond material. The electron directly crosses the FIB lamella (usually in the range of being 0.5 µm thick), and only the e-beam spot size excites the diamond material. Then, carrier transport gives the spatial resolution of the technique.

Other complementary analysis can be carried out by TEM to confirm the optical spectroscopy analysis. In this case, the experimental analysis becomes heavier as the sample preparation requires the fabrication of FIB lamella, similar to those used for CL, but much thinner (less than 100 nm thick). Indeed, diamond is so hard that it is nearly impossible to prepare lamella using traditionally used methods.

The most-used methodology in diamond materials is the lift-off FIB technique, which is summarized in Figure 4. A cross-sectional lamella is obtained for final polishing after trimming a thin diamond wall between previous deep trenches. Protection of the surface with some metal, as Pt, is required, especially if a close-to-surface interface needs to be studied. Final polishing up to some tens of nanometers is mandatory for certain TEM analyses. This final Ga^+^ ions etching can induce some amorphization or degradation of the crystalline structure if operation conditions are not well-controlled, which makes it unusable for HREM or HR-STEM studies.

The TEM analysis also allows assessing spectroscopic analysis as electron energy loss spectroscopy (EELS), where the bonding configurations can be evaluated. It is particularly powerful in the study of the B.C-N system, where the sp, sp^2^, or sp^3^ carbon hybrids can be evaluated [97]. The recently commercialized TEM microscopes can include a monochromator that leads to zero-loss peaks (ZLP) full width at half maximum (FWHM) in the range of 0, 1 eV. For spectroscopic analysis, this is an important technological improvement, and diamond is then an ideal material, due to its wide bandgap, to make spectroscopical analyses in the low loss range where the influence of the zero-loss peak [98] is now reduced. Damages, defect configuration, and changes in crystal structure attributed to B addition have been studied by EELS on polycrystalline diamond doping [99,100], where the variation of the C hybrids are well-revealed in the 300 eV range peak, which can also be related to the presence of point defects, dislocations, or vacancies. The technique is then complementary to CL, where such defects can also be observed. When HPHT is demonstrated to be able to deliver colorless diamond crystals (removing the usual brown color), the gem community and industry express great interest in understanding the origin of the different possible colors of diamond. Investigations allow attributing the blue to boron doping, the yellow to nitrogen doping, and the red to NV centers, and the brown is then related to the presence of vacancies clusters or nodes of dislocations [101]. However, this last aspect is still an open question.

In addition to carbon bonding configurations, EELS is also sensitive to the boron-binding states in the spectroscopical analysis and also allows the evaluation of the spatial distribution of boron dopant through the mapping facilities of the TEM, either used in the EFTEM mode (energy filtering TEM) or in the STEM (scanning TEM) one. This imaging mode of the TEM can also evidence distribution of B-B entities and point defects. Concerning dopants, due to their different sensitivity and spatial distribution, Raman is then also complementary to CL and EELS, as some vibrational modes of B allow the evaluation of the doping level of the diamond crystal [102].

As diamond has a relatively large bandgap with respect to the FWHM of the ZLP, valence electron energy loss spectroscopy in STEM (VEELS) allows the evaluation of the bandgap and the complex dielectric function with nanometers’ resolution. Special attention to the Cherenkov and Plasmon peaks has to be taken, and such experiments are recommended to be carried out at low electron beam energy (<80 keV, typically 60 keV) on relatively thin FIB lamella (<50 nm) [103]. The bandgap and dielectric constant of polycrystalline alumina onto diamond has been then estimated using this methodology [95].

Quantitative compositional information can also be obtained by high-angle annular dark field (HAADF) in the STEM mode [104,105]. Figure 5 shows a comparison between B-dopant determination by HAADF and SIMS. High-angle scattering of electrons is dominated by inelastic scattering, which means that no diffraction effects are produced if the collection angle is high enough so that the scattered intensity depends directly on the square of the atomic scattering factor. This has been demonstrated to be especially useful to quantify boron doping profiles with nanometric resolution in δ-doped layers [106,107] (down to 10^20^ at./cm^−3^ and 5 nm thick). This TEM related technique, however, is suitable to evaluate boron content when the dopant level is high, over the 10^19^ cm^−3^ scale. When boron level is below that, CL on cross section foils constitutes a more suitable methodology [108].

By far, the control of defects, especially dislocations, in diamond is one of the main goals for diamond researchers, as they can be responsible for future leakage currents in the device. Here again, TEM is a powerful tool to fully analyze the dislocation generation when growing CVD diamond so that some design rules can be offered to diamond community. It is very well-known from III–V semiconductors that a critical thickness on the epilayer is necessary to start dislocation formation as a consequence of an energy balance when plastic relaxation starts [40,109,110,111,112]. However, in the case of diamond, proximity effects concerning boron atoms in a closed diamond lattice have been shown to also be effective for that. Therefore, not only a critical thickness is taken into account, but a critical boron content can also be defined [38,39,113], which is of maximum interest to diamond growers. Introduction of other dopants, such as P, also induces defect formation, as can be seen in Figure 6 where a diffraction contrast TEM micrograph is shown. Plenty of dislocations and planar defects are formed in this phosphorous-doped diamond epilayer ([P] = 2.5 × 10^20^ at./cm^3^).

## 5. Diamond Electronic Devices

### 5.1. Schottky, Ohmic Contacts, and Diamond SBDs

Diamond Schottky barrier diodes (SBDs) have been extensively studied, and high breakdown voltage of >10 kV, high temperature operation, and low on-resistance have been reported [114,115,116,117,118]. Mostly, the p-type layer is used for the drift layer and contact layer, because the p-type layer is easier to control with doping concentration (10^15^–10^22^ cm^−3^) [119,120] and it shows higher Hall mobility of ~2000 cm^2^ V^−1^ s^−1^ [121] compared to n-type layers.

The quality of metal/diamond interfaces is one of the most important issues to obtain high performances in SBDs. Surface termination has an important role as well as other semiconductors, and it drastically changes the electrical characteristics of diamond surface. Generally, oxygen termination is adapted to perform stable Schottky contacts with higher Schottky barrier height (SBH). An acid mixture (e.g., H_2_SO_4_ + HNO_3_ at 200 °C), oxygen plasma treatment (or ashing), and exposure to ultraviolet (UV) under ozone atmosphere [25,122,123,124,125,126] are widely used to obtain O-terminated surfaces. Metals with a high-temperature melting point, such as Mo, Pt, Ru, and Zr, have resulted in high performances for high-voltage SBDs [127,128,129], although various metal species have been investigated for Schottky contacts [130,131,132,133,134,135,136]. SBH is reported to be 1.2–3.4 eV [28,137,138], depending on the metal species and surface treatments. In contrast, the n-type layer has high resistivity at room temperature due to the large donor activation energy (0.57 eV for P, 1.7 eV for N), and large SBH (4.3–4.5 eV) was found independent of metal species due to strong Fermi level pinning [33,139,140,141,142].

For ohmic contacts, titanium (Ti) is the most widely used for both p-type and n-type diamonds. Typically, ohmic electrodes are formed by depositing Ti (Ti/Au or Ti/Pt/Au) and annealing in N_2_ or Ar atmosphere. Ohmic characteristic is considered to be improved by a chemical reaction between Ti and diamond, such as carbide formation [128,143]. A low specific contact resistance of 2.8 × 10^−7^ Ωcm^2^ has been reported for p-type diamond (100)/Ti with annealing at 420 °C for 60 min in an Ar atmosphere [144].

Figure 7 shows schematic illustrations of diamond vertical-type SBDs proposed for power devices. These diodes have exhibited a breakdown voltage of 1.8–3.7 kV [115,136,145,146]. A maximum forward current of 10 A (electrode area of 16 mm^2^) has been reported for vertical SBD (VSBD) [147]. The electric breakdown field of 7.7 MV/cm has been published [117,129,148]. A reverse leakage current can be explained by thermionic field emission (TFE) + barrier lowering [149,150,151]. The abrupt leakage current increasing and the breakdown field lowering are suggested to be caused by defects in the diamond derived from substrate, CVD growth, and/or device processing [136,152,153,154], although effects of crystallographic defects have not yet been clarified. Metal-assisted termination (MAT) has been proposed as a buffer layer for CVD growth to reduce density of threading dislocation and to improve crystal quality and SBD properties [155].

### 5.2. Diamond pn and PiN Junction Devices

PiN diodes are expected to have high reverse-blocking voltage because a depletion region between the p-type layer and n-type layer (drift layer thickness for punch-through type) can support the electric field. Compared to SBD, it has higher on-voltage due to the bandgap energy and lower switching speed due to the longer recovery time of accumulated carriers in the drift layer. However, in an ultra-high-voltage region such as >10 kV, considerable lowering of specific on-resistance by conduction modulation can be an advantage while Si devices develop very high resistance due to the large thickness of the drift layer.

A high-quality diamond pn junction has been achieved by Koizumi et al. in 2001 [156], followed by a number of reports on optimized diamond pn or pin UV LEDs [157,158]. B-doping and P-doping are widely used for the p-type layer and n-type layer, respectively. Large activation energies of B-acceptor (0.37 eV) and P-donor (0.57 eV) cause high resistivity at room temperature. Nevertheless, recent studies showed that an extremely high doping level (10^22^ cm^−3^) is possible both for p-type and n-type while keeping high quality of crystal and junctions, which enables high carrier injection current under bipolar regime [159]. In addition, high P-concentration n-type layers on <100>-oriented devices have been achieved by overgrowth on shape-processed (100) diamond [160,161]. Kato et al. have reported on successful diamond bipolar junction transistors with this technique in the n-type layer [162].

Regarding the reverse blocking properties of diamond PiN diodes, a breakdown voltage (*BV*) of 11.5 kV with rectification ratio of 10^7^ has been reported [163], as shown in Figure 8. The breakdown was clear and non-destructive as shown. The diode structure was fabricated by MPCVD growth of undoped (intrinsic) and P-doped n-type homoepitaxial layers on an HPHT (100) IIb p^+^-type diamond substrate followed by forming Ti/Pt/Au electrodes on both top and bottom surfaces [164]. The thickness of the drift layer (undoped layer) is a 70 μm thick drift layer, and the corresponding breakdown field (*F_B_*) is estimated to be 1.9 MV/cm with assuming a punch-through state by:(1)$$\left|BV\right|=\left|F_{B}\right|-\frac{qN_{A}d^{2}}{2\varepsilon_{S}}$$
where *d* is the drift layer thickness, *q* is the electronic charge, *N_A_* is the acceptor concentration of the drift layer, and *ε_S_* is permittivity. Figure 9 shows a comparison of reverse I–V characteristics between diodes with a mesa structure and without a mesa structure for the diamond PiN diodes (36 μm thick drift layer) [163]. It is found that the reverse leakage current is considerably reduced by the mesa structure. This result suggests that the reverse leakage current in the diodes can be passed through the n-type layer or surface. The *BV* and *F_B_* of diamond pin diodes are shown in Figure 10 as a function of the drift layer thickness. The *BV* increased with the increase the drift layer thickness, closely tracking a theoretical calculation result [165], although the value is somewhat smaller than that. The maximum value of *F_B_*, 3.6 MV/cm, was obtained for the PiN diode with 2 μm thick drift layer [163]. This value is higher than that of theoretically predicted values of GaN or SiC. Higher *F_B_*, such as >10 MV/cm, should be realized by proper terminations, device structures, and higher crystal quality.

Other diodes with pn junctions have been also proposed for high-power devices, as shown in Figure 11. The Schottky–pn diode (SPND) is tandemly merged SBD with a pn diode (PND) [166] (Figure 11a). This diode is a unipolar device that shows lower on-voltage than that of PND and lower specific on-resistance and higher reverse blocking properties compared to SBD. Forward current density of 60 kA/cm^2^ at 6 V (corresponding R_on_S = 0.03 mΩcm^2^) with rectification ratio of 10^12^ and 3.4 MV/cm in a reverse blocking field has been reported for SPND [167]. Schottky PiN diodes have also been demonstrated with a blocking voltage of 500 V [168]. A high-voltage vacuum-power switch with a diamond PiN diode has been also proposed for the ultra-high-voltage region (Figure 11b). This device is utilizing highly efficient electron emission from the diamond PiN diode based on the negative electron affinity (NEA) of diamond. Takeuchi at al. demonstrated a 10 kV vacuum switch with high-power transmission efficiency of 73% using a diamond PiN diode [169].

### 5.3. Diamond FETs

Diamond field effect transistors (FETs) have been also widely studied for high-power and/or high-frequency switching devices, as shown in Figure 12. Diamond metal semiconductor FETs (MESFETs) with a p-type Schottky junction gate (Figure 12a) have exhibited the breakdown voltage of >2 kV [170]. High temperature operation and high radiation tolerance have been reported [171]. Junction FETs (JFETs) are expected to be highly reliable for high-temperature and high-voltage operation because of the pn junctions instead of a gate oxide (Figure 12b). Normally off diamond JFETs with a high current density of 458 A/cm^2^ have been achieved [172].

Inversion metal-oxide semiconductor FETs (MOSFETs) are one of the most widely used electron devices. High-quality MOS interfaces have been difficult to fabricate on diamond due to the lack of natural oxide layers. Recently, thanks to improvements in the MOS interface by O-H termination with a wet-annealing technique and a higher quality of n-type layer, a diamond inversion-type p-channel MOSFET with normally-off operation has been realized [173] (Figure 12d). The field-effect (inversion channel) mobility has been estimated to be 8 cm^2^ V^−1^ s^−1^, and the low mobility can be caused the existence of a high interface state density of 6 × 10^12^ cm^−2^ eV^−1^. The field-effect mobility was found to be dependent on the interface state density, which increased with the increase in the roughness of the Al_2_O_3_/n-diamond interface at the channel [174]. The roughness increased with the increase in the phosphorus concentration in the n-layer, and the improved field-effect mobility of 20 cm^2^ V^−1^ s^−1^ has been obtained by reducing the interface state density [174]. Moreover, for bulk FETs, recently, deep depletion MOSFET has been proposed and demonstrated [175]. A breakdown field of 4 MV/cm has been obtained for a lateral normally on device consisting of oxygen-terminated diamond [176].

H-terminated FETs have been also extensively investigated. Diamond has the unique property that, near the hydrogen-terminated (C-H) surface, high-density hole accumulation (~10^13^ cm^−2^) forms two-dimensional hole gas (2DHG) with very low activation energy and high hole mobility [25,177,178]. Accordingly, hydrogen-terminated diamond FETs with 2DHG show high transistor performances from the point of current density, high transconductance, and high-frequency operation. Both diamond MESFET and MOSFET with 2DHG have exhibited high-frequency operation (GHz) since 2001 [179,180,181,182]. These FETs are promising for application to high-power radio-frequency (RF) power amplifiers beyond other wide bandgap semiconductor devices. A cut-off frequency (f_T_) of 70 GHz^182^ and a maximum oscillation frequency (f_max_) of 120 GHz [180] have been achieved. These values are comparable to GaN-based HEMTs [183,184,185]. The maximum drain current density (I_D max_) of 1.35 A/mm [186], the blocking voltage of >2 kV for normally off operation devices [187], and the microwave output power (P_out_) of 3.8 W/mm at 1 GHz [188] and 1.5 W/mm at 3.6 GHz [189] have been also reported. Improvement in sheet resistance and contact resistance can provide further improvement in output power [190]. Recently, vertical (trench gate structure) MOSFETs with side wall 2DHG are also proposed and demonstrated [191,192], which have exhibited maximum drain current density of 710 mA/mm. In addition, the above-mentioned diamond H-terminated 2DHG FETs, which have a p-channel, can be highly promising for complimentary circuits with GaN n-channel FETs in power amplifiers.

In this section, the current status of diamond electronic devices has been reviewed, focusing on power semiconductor devices. Diamond devices have been remarkably improved thanks to the establishment of MPCVD growth techniques including doping control and characterization techniques. In this decade, several diamond devices have exhibited excellent performances beyond other semiconductors based on the material advantage. However, the full potential of remarkable advantage of diamond has not yet been demonstrated. One of the big issues is that inadequate device fabrication techniques are limiting device performances. Etching technique, interfaces in MOS, ion implantation, or selective doping (for edge termination, buried structure, and so on), and passivation materials are key techniques to obtain higher performances. Furthermore, for ultra-high-power electronics applications, bipolar devices should be necessary, such as GTO, thyristor, and IGBT in addition to PiN diodes in which sophisticated device fabrication techniques and doping techniques are crucial. Due to a great deal of effort, techniques of selective growth [193,194], selective doping [63,72,160,195,196], and selective etching [161,197] have made great progress in fabricating device structures. The problems of n-type layers are still not fully solved. However, today, n-type doping level is possible to control from 10^15^–10^20^ cm^−3^, and then significant reduction in resistivity by using hopping conduction has been reported in the pin structure [149], and improvement in the crystal quality of phosphorus-doped (n-type) diamond has been reported [67,174]. As a summary, diamond devices still have many issues for practical use; however, considering the remarkable development in recent years, ultra-high-power diamond devices can be achieved both for HVDC and RF applications in the near future.

## 6. Summary and Conclusions

This paper gives a brief overview of the state of the art of the diamond technology for power devices. It shows that strong limitations of the material for such applications have now been overcome. However, its high hardness and epitaxial growth out of equilibrium are not straightforwardly resolved and have taken some decades. Example in the characterization method and the growth of p-type and n-type diamond show the high potential of this material. High-quality vertical and lateral growth have been demonstrated, and the example of Schottky and p–n diodes as well as transistors are shown to work under high-power electronic conditions. This shows that the diamond is competitive with other WBG materials for the power electronic niche. Several device architectures have been recently manufactured, even though reliability is still a pending subject for it commercialization. Therefore, the material is close to finding some application as an electronic material, making it possible to take advantage of its outstanding electronic properties.

## Figures and Tables

**Figure 1 materials-14-07081-f001:**
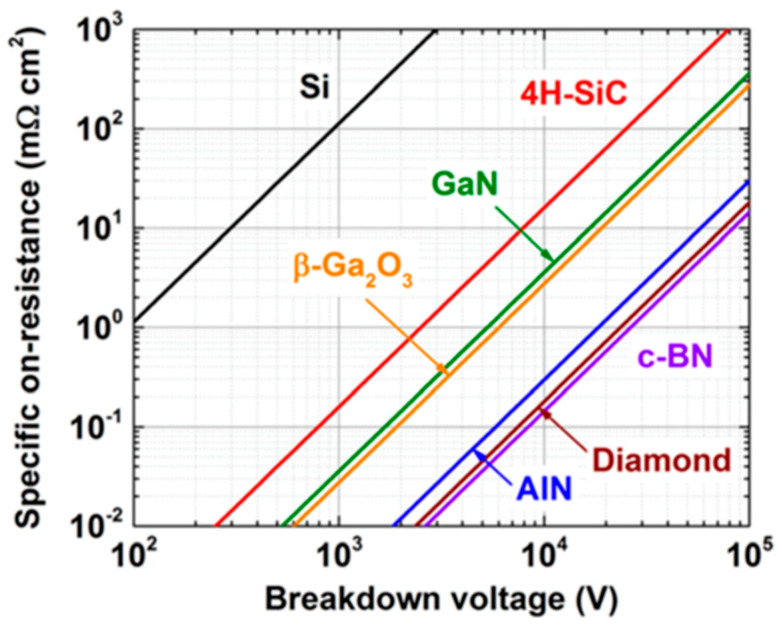
ON-resistance and breakdown voltage of the different semiconductors at room temperature. Note that at high temperature diamond improves its characteristics (Reproduced with permission from Pearton et al. [5] Copyright 2021 ©AIP Publishing).

**Figure 2 materials-14-07081-f002:**
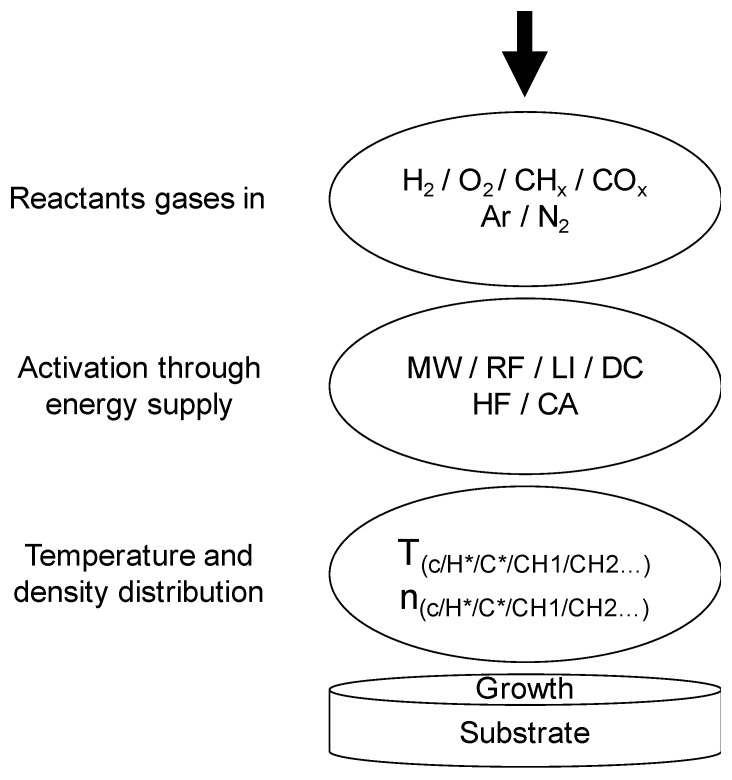
Schematic diagram of the mechanism for diamond CVD growth processes. The acronyms used in the figure correspond to: MW, microwave; RF, radiofrequency; LI, laser induced; HF, hot filament; DC, direct current; CA, chemical activation.

**Figure 3 materials-14-07081-f003:**
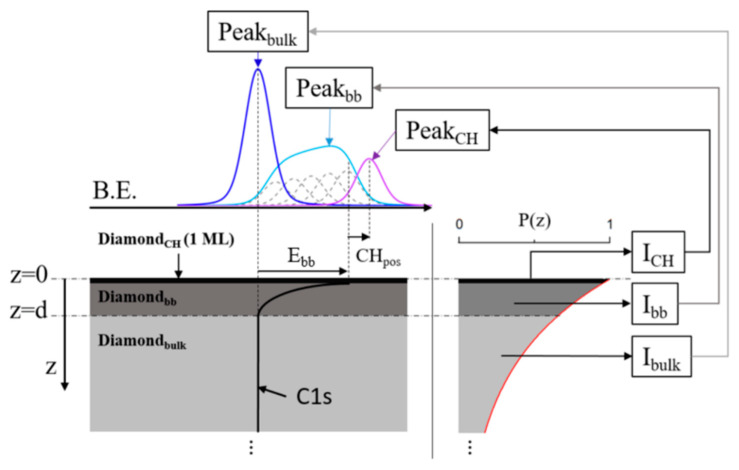
Schematic of the XPS model used for C 1s peak analysis. Sample is divided into three regions: diamond bulk (Diamond_bulk_), diamond with band bending (Diamond_bb_), and diamond surface C-H_x_ (Diamond_CH_) for the first monolayer of material. The intensities I_bulk_, I_bb_, and I_CH_ of the respective XPS peaks Peak_bulk_, Peak_bb_, and Peak_CH_ are obtained from their respective ratios. Peak_bulk_ and Peak_CH_ follow Voigt distributions; Peak_bb_ is defined by its band bending width (d) and band bending (E_bb_). The latter is set to the Peak_bulk_ position. Adapted from the graphical abstract of Alba et al. [83].

**Figure 4 materials-14-07081-f004:**
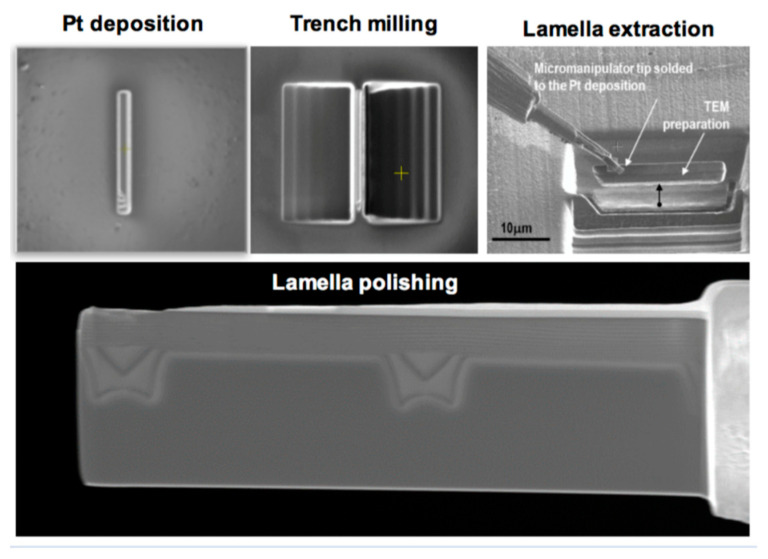
Lift-off methodology for the preparation of TEM lamella in diamond materials. After Pt deposition for surface protection, two parallel trenches are milled at both sides, and, after cutting off and removing using a micromanipulator tip, final polishing is performed.

**Figure 5 materials-14-07081-f005:**
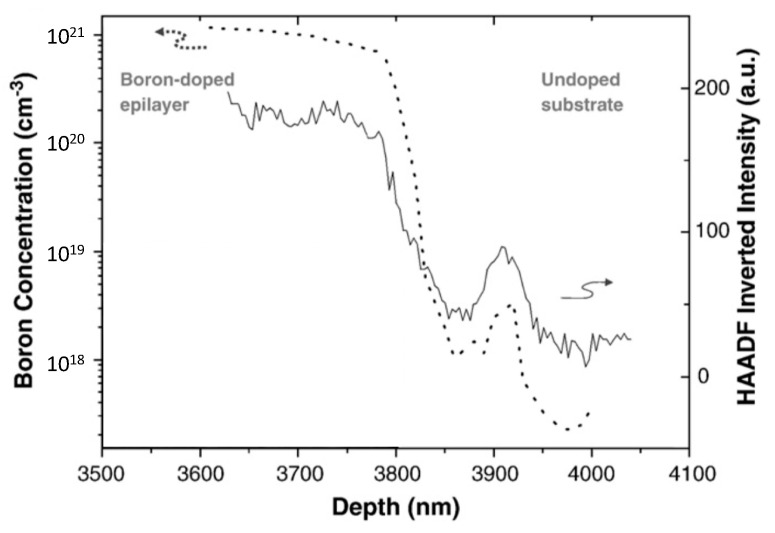
A comparison between HAADF and SIMS is shown for B determination in a boron-doped diamond homoepitaxial layer. The dashed curve corresponds to the SIMS profile and the continuous line to the inverted HAADF profile. Boron can be clearly detected in the 10^20^ cm^−3^ range. Adapted with permission from Araujo et al. [104] Copyright 2021 ©Elsevier.

**Figure 6 materials-14-07081-f006:**
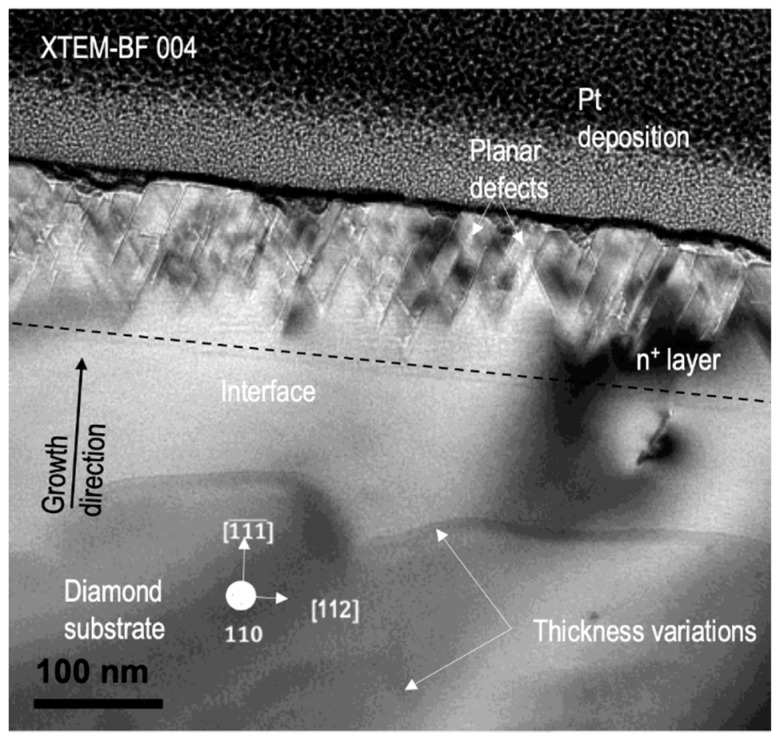
Diffraction contrast TEM micrograph, recorded with the 004 reflection, shows the formation of dislocations and planar defects above the interface for a heavily P-doped diamond epilayer.

**Figure 7 materials-14-07081-f007:**
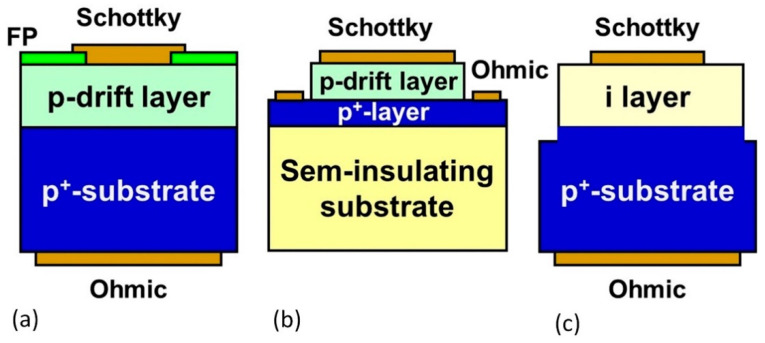
Schematic illustrations of vertical-type diamond Schottky barrier diodes. (**a**) Vertical SBD. (**b**) Pseudo-vertical SBD. (**c**) Metal-intrinsic SBD.

**Figure 8 materials-14-07081-f008:**
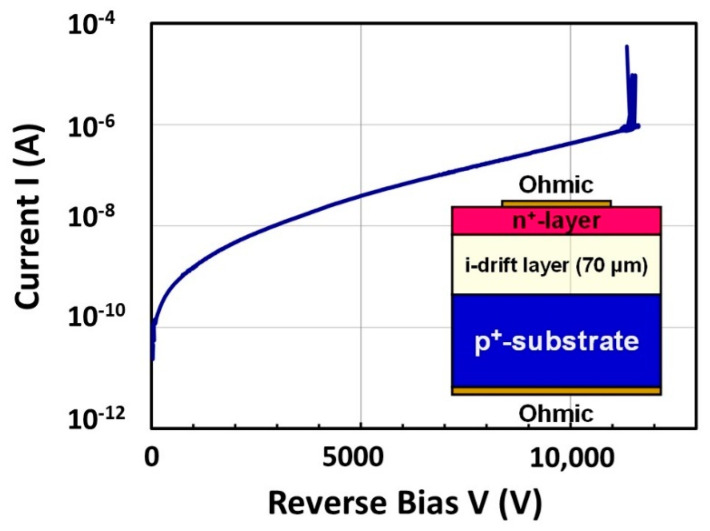
A representative reverse I–V property for diamond PiN diode (drift layer thickness 70 μm).

**Figure 9 materials-14-07081-f009:**
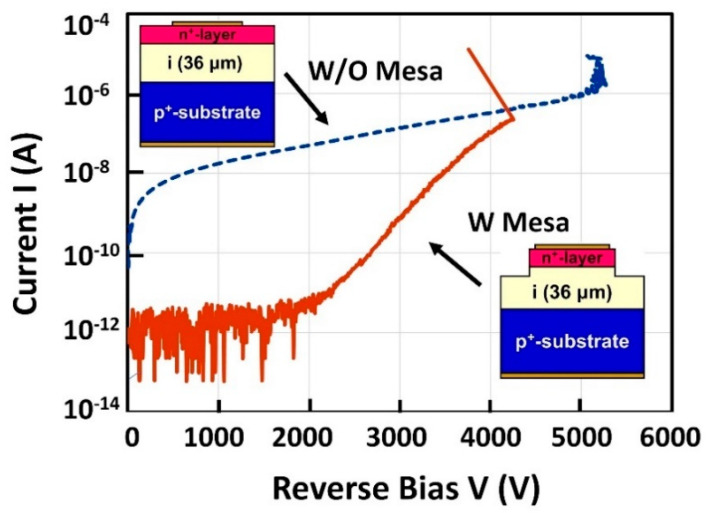
Reverse I–V properties for diamond PiN diode both with mesa structure and without mesa structure (drift layer thickness 36 μm).

**Figure 10 materials-14-07081-f010:**
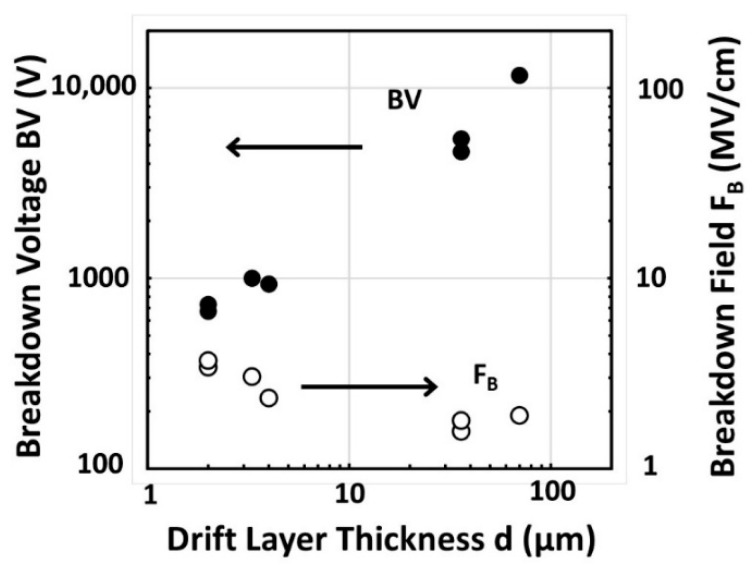
Break down voltage and breakdown field of diamond PiN diodes as a function of the drift layer thickness.

**Figure 11 materials-14-07081-f011:**
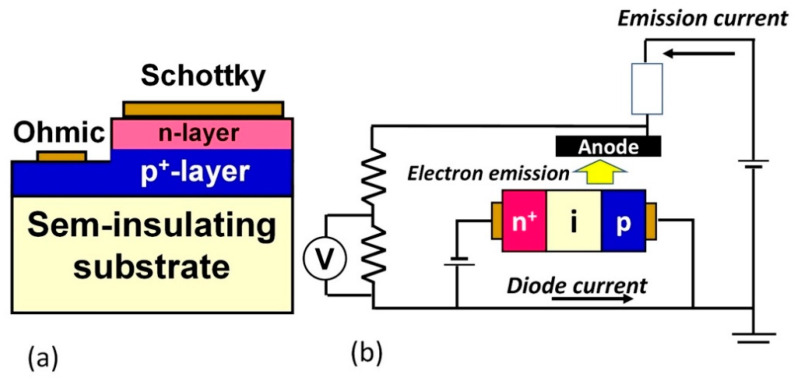
Schematic illustrations of noble diamond diodes. (**a**) Schottky pn diode (SPND). (**b**) Vacuum switch utilizing highly efficient electron emission from diamond PiN diode.

**Figure 12 materials-14-07081-f012:**
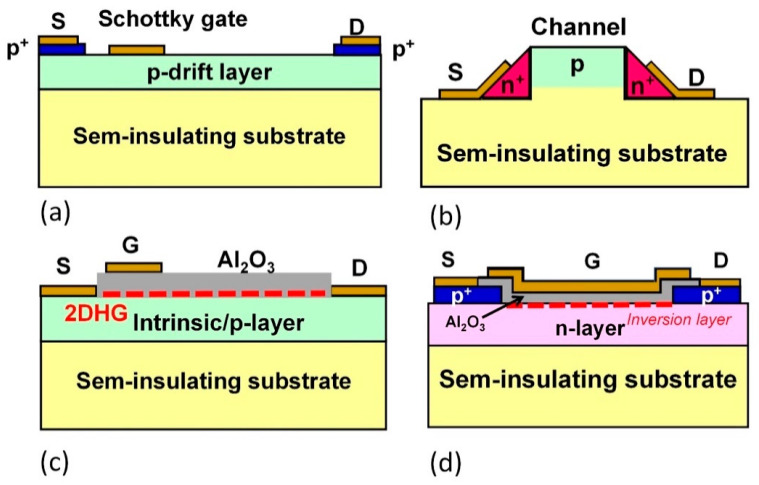
Schematic illustrations of diamond FETs. (**a**) MESFET. (**b**) Lateral pn junction JFET. (**c**) H-FET (C-H 2DHG MOSFET). (**d**) Inversion MOSFET.

**Table 1 materials-14-07081-t001:** Diamond’s properties are strikingly superior to other semiconductors (see [5]) when considered for use in power electronic devices. Johnson’s figure of merit (FoM) is a measure of the ultimate performances of the electronic device at high power and high frequency of a transistor, Keyes’ FoM measures the performance limited by heat generation and removal, and Baliga’s FoM measures performance limited by losses at high-power and high-frequency operation. The properties/figure where diamond is outstanding for the present project are highlighted in orange.

Property (Unit), (See [5])	Si	SiC-4H	GaN	Ga_2_O_3_	Diamond
Bandgap (eV)	1.1	3.23	3.42	4.8	5.45
Dielectric constant, ε	11.8	9.7	9	10	5.7
Breakdown field (MV/cm)	0.3	3	2	8	10
Electron mobility (cm^2^/Vs)	1500	1000	2000	300	1000
Hole mobility (cm^2^/Vs)	480	100	20		2000
Thermal conductivity (W/cmK)	1.5	5	1.5	0.27	22
Johnson’s figure of merit (10^23^ ΩW/s^2^)	2.3	900	490	1236	2530
Keyes’ figure of merit (10^7^ W/Ks)	10	53	17	2	218
Baliga’s figure of merit (Si = 1)	1	554	188	3214	23,068

## Data Availability

The data that support the findings of this study are available from the corresponding author upon reasonable request.

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
