# Peer review of "Diamond for Electronics: Materials, Processing and Devices"

_materials, 2021, doi:10.3390/ma14227081_

Round 1

Reviewer 1 Report

This article provides an overview of the current knowledge regarding the use of diamonds in electronics.  Autors will report the recent key issues related to crystal growth, characterisation techniques and, in particular, the importance of surface  states aspects, fabrication processes and device fabrication. Finally, the advantages and disadvantages of diamond devices respect to other WBG materials are also discussed.

On this basis the manuscript could be accepted after minor modification.

Line 139  ... ®  ...    typos?

Line 140 two dots

Line 163 Please explain in figure 2 the abbreviations MW / RF / LI / DC / HF? CA

Line 163 What does lowercase c mean in parentheses next to T and n?

some typos in text  

Author Response

Thank you very much  for the comments that allowed to improve the manuscript. 

The text has been improved regarding the reviewer comments. In Fig.2, the acronyms have been explained in the figure caption.

Reviewer 2 Report

  1. The affiliations of all the authors are not listed.
  2. The dielectric constant of diamond should be 5.7,not a comma. And the electron and hole mobilities are 4500 and 3800, respectively. Many references are indicated this, such as “Science, vol. 297, no. 5587, pp. 1670–1672, Sep. 2002, doi: 10.1126/science.1074374”.
  3. Reference 36 and 54 are identical.
  4. Reference 73 and 76 should be the year of 2021.
  5. In page 7, line 227, the first sentence mentioned the “doping level”, however, the rest of the paragraph talked about the dislocation instead of this issue.
  6. The application of doping in power electronics should be more illustrated. Since without well doping profiles, the complex device structures, such as IGBT, VDMOS, GTO, can not be realized in diamond devices.
  7. In page 16, line 513, the unit of breakdown field should be MV/cm.
  8. For the 2DHG FET, maybe you should mention some high frequency reports, such as 120GHz fm and 70 GHz ft. One of the limited effects is the actual low carrier mobility of 2DHG now. If it’s as high as 2DEG in GaN/AlGaN, the diamond FET can be competitive.

Author Response

Tank you very much to the reviewer for these remarks that allowed us to improve the manuscript. 

In attached a brief summary of the modifications we introduced regarding your remarks:

Reviewer 3 Report

In this paper, authors gave a very complete review on the diamond eletronics, including the materials, processing and devices.
However,there are still some points to be improved. 
1. Generally ,it is accepted that hole mobility of diamond is 1800cm2/Vs,and electron mobility is 2200cm2/Vs.
2. "Indeed, most of the reported high-quality diamond (100) films were grown with Cme below 1.0%. Consequently, the typical Rg is lower than 1 mm/h."
In fact, the Rg is generally several or several tens of um/h. "lower than 1 mm/h" seems not reasonable. 
3. Authors summarized the characterization methods of defects. If for the trace impuries, such as B or N concentration less than 100ppb, which method will be ultilized?
4. This paper focuses on the power devices. So more power performances on the diamond devices such as H-terminated diamond should be added.
5. Surface roughness is an important parameter for the devices. Please also give some contents on this. 

Author Response

We thank the reviewer for the comments that allowed us to improve the manuscript. In the attached document, the response point by point (in red) of the remarks and comments.
